# Osteoporosis Assessment among Adults with Liver Cirrhosis

**DOI:** 10.3390/jcm12010153

**Published:** 2022-12-25

**Authors:** Claudiu Marinel Ionele, Adina Turcu-Stiolica, Mihaela Simona Subtirelu, Bogdan Silviu Ungureanu, Teodor Nicusor Sas, Ion Rogoveanu

**Affiliations:** 1Doctoral School, University of Medicine and Pharmacy of Craiova, 200349 Craiova, Romania; 2Department of Pharmacoeconomics, University of Medicine and Pharmacy of Craiova, 200349 Craiova, Romania; 3Research Center of Gastroenterology and Hepatology, University of Medicine and Pharmacy of Craiova, 200638 Craiova, Romania; 4Department of Gastroenterology, University of Medicine and Pharmacy of Craiova, 200349 Craiova, Romania; 5Department of Radiology and Imaging, University of Medicine and Pharmacy of Craiova, 200349 Craiova, Romania

**Keywords:** cirrhosis, chronic liver disease, bone mineral density, osteoporosis, Controlling Nutritional Status index, nutritional status

## Abstract

Osteopenic bone disease occurs frequently in patients with chronic liver cirrhosis, which most frequently presents with hepatic osteodystrophy. Thus, the relationship between nutritional status and bone mineral density has been poorly measured in liver cirrhosis. This single-center study consisted of a group of 70 patients diagnosed with liver cirrhosis. The nutritional status was evaluated with the Controlling Nutritional Status index, and volumetric vertebral bone mineral density was measured with quantitative computed tomography. Among the 70 patients included, osteopenia and osteoporosis were found in 71% and 24.3%, respectively. Malnutrition assessed with the Controlling Nutritional Status index was observed in 56 (80%) patients and was more frequent in alcoholic cirrhosis patients than viral cirrhosis patients (87.24% vs. 65.22%). Significant positive correlation with Controlling Nutritional Status score was found with Model for End-Stage Liver Disease (rho = 0.576, *p*-value < 0.0001), Child–Pugh score (rho = 0.670, *p*-value < 0.0001), International Normalized Ratio (rho = 0.517, *p*-value = 0.001), aspartate aminotransferase (rho = 0.293, *p*-value = 0.045), and bilirubin (rho =0.395, *p*-value = 0.02). Among the liver cirrhosis patients, 15 had osteoporosis and 49 had osteopenia at the lumbar spine (L1-L4 vertebrae), as determined by bone mass density via quantitative computed tomography. A non-significant relationship between Controlling Nutritional Status index-assessed nutritional status and bone mass density was documented. Regarding osteoporosis, no differences were found between the viral and alcohol types of liver cirrhosis patients (*p*-value = 0.870). Age, obesity, grade of varices, Child–Pugh score, and Model for End-Stage Liver Disease score were associated with osteoporosis in patients with liver cirrhosis.

## 1. Introduction

The liver is considered an important center for a multitude of physiological processes [1]. It is seen as the main organ for metabolizing three major classes of molecules (protein, fat, also carbohydrate) [2]. Liver processes consist of macronutrient metabolism, breakdown of xenobiotic compounds, also a large number of current drugs, immune system support, blood volume adjusting, endocrine control of growth signaling pathways, and lipid and cholesterol homeostasis [1,3]. Liver cirrhosis (LC) is a normal outcome of all chronic liver diseases (CLD) and can be characterized by tissue fibrosis but also by a transformation of the normal liver framework into structurally abnormal nodules [4]. It may develop over a period of time through liver chronic inflammation and is acknowledged as the end-stage form of CLD that can be continued with a number of nutrition disorders [5]. Among these, protein-energy malnutrition (PEM) can be seen as a common complication in LC patients, which also is linked with a high risk of morbidity and mortality [6,7]. Therefore, accurate nutritional assessment is fundamental for LC patients’ management. Among the main causes of LC, viral hepatitis and alcohol consumption are two of the most important ones [8,9,10]. 

Osteoporosis is defined by the World Health Organization (WHO) as being a systemic skeletal disease mostly defined by a downturn in bone mineral density (BMD); this may lead to an increased morbidity and mortality due to fractures [11]. First of all, bone densitometry (dual-energy X-ray absorptiometry, DEXA) was considered the right method for the measurement of bone density and was used for diagnosing osteoporosis in LC. More recently, newer methods such as quantitative computed tomography (QCT) were successfully applied for the assessment of BMD [11,12,13,14]. The American College of Radiology describes osteoporosis in QCT as a BMD under 80 mg/cm^3^. For osteopenia, the BMD ranges between 80 and 120 mg/cm^3^, and a value higher than 120 mg/cm^3^ defines a normal bone density. The categories presented were derived by selecting thresholds that result in roughly the same proportion of the population being assigned to a specific category based on QCT spine T-score [15]. 

The Controlling Nutritional Status (CONUT) score is considered an unbiassed tool that is utilized to assess the nutritional status in different disease stages [6,16,17]. Thus, the CONUT score is viewed as an index calculated from three parameters: total cholesterol level, serum albumin value, and peripheral lymphocyte count, which are descriptive markers of protein synthesis, caloric shortcomings, and impaired immune defenses [18,19]. Initially, the CONUT index was proposed and used as an instrument for the early stages of detection of hospitalized patients with poor nutritional status, but more recently, it is considered to be a good prognostic index for long-term follow-up in patients experiencing CLD, with malignant tumors such as hepatocellular carcinoma (HCC), colonic cancer or pancreatic ductal adenocarcinoma [16,18,20,21]. 

Hence, we carried out a study having as goal the identification of predicting factors for osteoporosis in LC patients using both laboratory parameters and imaging data obtained from opportunistic QCT.

## 2. Materials and Methods

### 2.1. Patient Selection

Newly diagnosed patients with LC hospitalized in the gastroenterology department of the County Clinical Emergency Hospital of Craiova, Romania, were selected for our analysis. The inclusion period was between January 2019 and December 2020. We included patients diagnosed with viral B-induced LC, viral C-induced LC, and alcohol-consumption-related disease [22]. All patients had opportunistic QCT (i.e., computed tomography—CT acquired for other medical reasons). Exclusion criteria were (1) patients with hepatocellular carcinoma or other malignancies, autoimmune liver diseases, kidney disease, chronic gastritis [23]; (2) patients treated with oral supplements containing magnesium, calcium, vitamin D, phosphorus, diuretics, and tenofovir disoproxil [24]. If fractures, cement-augmented vertebral bodies, anatomical deformities, or implants were detected in the CT scan, the respective patients were not included in the measurements. 

The present study protocol followed all regulations of the Declaration of Helsinki and was approved by the ethics committee of the University of Medicine and Pharmacy of Craiova, Romania (no. 173/29 October 2021).

### 2.2. Imaging

Measurements were performed using computed tomography (CT) and we had a General Electric Light Speed Series, with a helical 64-channel, Revolution, and a Siemens Biograph mCT 20 slices. Scanner settings of 130 kVp were used. CT images were retrospectively analyzed using a largely advanced medical imaging service for picture archiving and communication (Biotronics 3Dnet PACS). A dual-energy lumbar X-ray absorptiometry (DEXA) was performed on Hologic systems Horizon A (S/N200639). The reconstructions were obtained in the axial plane (Figure 1A) of the lumbar vertebrae from L1 through L4. The region of interest (ROI) was chosen to be around 20 mm, leaving out the cortex (Figure 1B). The CT attenuation was measured in Hounsfield units (HU) by sketching a click-and-drag elliptical ROI within the axial section of the lumbar trabecular bone. ROI avoided degenerative changes to the vertebrobasilar complex and cortical surfaces. The following formulas were used to approximate bone density values for the General Electric CT scan: QCT-value = 0.71 × HU + 13.82 mg/cm^3^, and, for the Siemens CT scan: QCT-value = 0.985 × HU + 15.516 mg/cm^3^ [12,25]. 

### 2.3. Biological Analyses

Fasting blood samples were taken for measuring the international normalized ratio (INR), quick time (TQ), prothrombin time (TP) aspartate aminotransferase (AST), alanine aminotransferase (ALT), gamma-glutamyltransferase (yGT), phosphatase, calcium (mmol/L), platelet count, hemoglobin, creatinine, and total bilirubin, with all of them being determined by automated routine procedures.

We used Child–Pugh and MELD scores to evaluate prognosis in LC. We also determined the MELD score for all the patients, which is based on total bilirubin, creatinine, and INR. The Child–Pugh scoring system was also assessed for all patients. This index uses six clinical and laboratory criteria to categorize patients: ascites, hepatic encephalopathy, nutritional status, total bilirubin, albumin, and INR. All of the selected patients were divided by Child–Pugh classification: A: from 5 to 6 points, B: from 7 to 9 points, and C: from 10 to 15 points, and the survival of cirrhotic patients was reduced if the Child–Pugh scores/classes were increased [26,27,28].

### 2.4. Intra-Rater and Inter-Rater Reliability

To perform intra-observer precision, the QCT measurements were carried out twice by the same author (C.M.I.) blinded to any information at 2-week intervals on a random sample of 20 patients. For inter-observer reliability, the QCT measurements were performed by two readers (C.M.I. and T.N.S) for L1 to L4 on a random sample of 20 patients. The assessed intraclass correlation coefficients (ICC) were considered excellent for values greater than 0.90 and good for values between 0.75 and 0.90 [29]. We also calculated internal consistency by Cronbach’s alpha (0.9: excellent; 0.8–0.9: good; 0.7–0.8: acceptable) [30]. 

### 2.5. Statistical Analysis

Besides descriptive statistics, different variables were compared using the Mann–Whitney U test or *t*-test (after checking the normality of continuous variables) or chi-squared test (for categorical variables). Moreover, various correlations between variables were calculated by using Spearman coefficients and visually presented with a correlation heatmap (colors range from bright blue for strong positive correlations, to bright green, for strong negative correlations). Missing data (the case for calcium and T-score) were the type that were missing completely at random due to the retrospective type of study, and we used mean substitution in order to reduce the standard error [31]. We applied univariable and multivariable regression analyses with the stepwise method to investigate independent associations between BMD as determined by QCT (as dependent variable) and other clinical and laboratory parameters (as independent variables). The standardized linear coefficients β (+95% CI) showing linear correlations between two parameters were determined. Statistical analysis was performed with Python (version 3.10.7) at a 0.05 level of significance (two-sided).

## 3. Results

### 3.1. Intra-Rater and Inter-Rater Reliability

For intra-observer precision, the ICC with Cronbach’s alpha of the QCT measurement was 0.955 (95% CI, 0.841–0.988) with 0.953 for L1, 0.942 (95% CI, 0.797–0.984) with 0.942 for L2, 0.972 (95% CI, 0.901–0.992) with 0.969 for L3, and 0.940 (95% CI, 0.787–0.984) with 0.934 for L4. Meanwhile, the ICC with the Cronbach’s alpha of QCT measurement for inter-observer precision was 0.943 (95% CI, 0.80–0.985) with 0.948 for L1, 0.946 (95% CI, 0.810–0.985) with 0.943 for L2, 0.975 (95% CI, 0.913–0.993) with 0.974 for L3, and 0.975 (95% CI, 0.913–0.993) with 0.976 for L4. The ICC’s values indicated an excellent level of agreement, and the Cronbach’s alphas demonstrated excellent internal consistency of the QCT measurements.

### 3.2. Patient Characteristics

Baseline data in all patients are presented in Table 1. During the study period, CT was performed for 70 patients (65 males in the majority, 92.9%, and 5 females, 7.1%). The age ranged from 53 to 66.5 years, and the patients with viral cirrhosis were significantly older than patients with alcohol cirrhosis (*p*-value = 0.020). There were 55 cases (78.6%) of history of hepatic decompensation, with more cases for patients with alcoholic cirrhosis (*p*-value = 0.015). Twenty-four cases (34.3%) had jaundice, with more cases for patients with alcoholic cirrhosis (*p*-value = 0.008). Of the 70 patients, 33 (47.1%) had cirrhosis complicated by ascites (no differences between the two groups, *p*-value = 0.247), 59 (84.3%) had varices (no differences between the two groups, *p*-value = 0.096), and 14 (20%) had hepatic encephalopathy (no differences between the two groups, *p*-value = 0.087). There were 25 (35.7%) patients in Child–Pugh A, 26 (37.1%) patients in Child–Pugh B, and 19 (27.1%) patients in Child–Pugh C (with more viral cirrhosis patients in Child–Pugh A and more alcohol cirrhosis patients in Child–Pugh B or C). The CONUT score has values between 0 to 11 (with median 5), according to which a normal nutritional state was found in 14 (20%) patients, a mild nutritional state in 18 (25.7%), a moderate nutritional state in 25 (35.7%), and a severe malnutrition state in 13 (18.6%) (no notable differences were found between the two groups, *p*-value = 0.158). Only 30 (43%) were obese or overweight.

Among the LC patients, 15 had osteoporosis and 49 had osteopenia at the lumbar spine (L1–L4 vertebrae), as determined by BMD via QCT. Regarding osteoporosis, no differences were found between viral and alcohol type of LC patients (*p*-value = 0.870), as shown in Table 2.

No differences were obtained for calcium levels between patients with and without osteoporosis (*p*-value = 0.493). Significantly higher values were obtained for QCT score (100.17 ± 16.16 vs. 70.42 ± 9.39, *p*-value < 0.001) and T-score (−1.61 ± 0.57 vs. −1.94 ± 0.73, *p*-value = 0.035) for patients without osteoporosis, comparative to the patients with osteoporosis.

Comparing the 25 patients in Child–Pugh A, 26 patients in Child–Pugh B, and 19 patients in Child–Pugh C, the CONUT score was significantly different, as in Figure 2: Child–Pugh A vs. B, *p*-value < 0.0001; Child–Pugh A vs. C, *p*-value < 0.0001; Child–Pugh B vs. C, *p*-value < 0.0001. The CONUT score was from 0 to 7 in patients with Child–Pugh A (mean ± SD, 2.28 ± 2.01; median (IQR), 2 (1–3.5)), 1 to 10 in patients with Child–Pugh B (mean ± SD, 4.96 ± 2.75; median (IQR), 5 (2–6.5)), and 2 to 11 in patients with Child–Pugh C (mean ± SD, 8.11 ± 2.21; median (IQR), 8 (7–10)).

The BMD derived from CT was not significantly different among the Child–Pugh classifications (Child–Pugh A vs. B *p*-values = 0.486; Child–Pugh A vs. C *p*-values = 0.515; Child–Pugh B vs. C *p*-values = 0.973), as in Figure 3. The BMD score ranged from 42.07 to 153.79 in patients with Child–Pugh A (mean ± SD, 96.26 ± 23.1; median (IQR), 93.83 (81.2–113.49)), 65.99 to 152.56 in patients with Child–Pugh B (mean ± SD, 92.81 ± 17.89; median (IQR), 90.87 (82.77–101.93)), and 59.88 to 124.2 in patients with Child–Pugh C (mean ± SD, 91.89 ± 16.18; median (IQR), 95.72 (77.82–102.15)).

Significant positive correlation with CONUT score was found with the Model for End-Stage Liver Disease index (MELD) (rho = 0.576, *p*-value < 0.0001), Child–Pugh (rho = 0.670, *p*-value < 0.0001), INR (rho = 0.517, *p*-value = 0.001), AST (rho = 0.293, *p*-value = 0.045), and bilirubin (rho = 0.395, *p*-value = 0.02), as shown in Figure 4. We obtained a positive significant correlation between the T-score (BMD DEXA) and lumbar QCT (Spearman’s rho = 0.361, *p*-value = 0.002).

The patients with jaundice had a CONUT score much higher than the patients without jaundice (*p*-value < 0.0001). The same significantly higher values were obtained in the case of the patients with encephalopathy (*p*-value < 0.0001), ascites (*p*-value < 0.0001), or history of hepatic decompensation (*p*-value < 0.0001).

Significant negative correlation with CONUT score was found with TP (rho = −0.284, *p*-value = 0.017) and hemoglobin (rho = −0.645, *p*-value < 0.0001).

CONUT score did not correlate with BMD determined by CT (rho = 0.078, *p*-value = 0.520).

The BMD determined by QCT was negatively correlated with age (rho = −0.645, *p*-value < 0.0001) and positively correlated with the varices grade (rho = 0.260, *p*-value = 0.030). Osteoporosis (small values of BMD) was found more significantly in patients with obesity (*p*-value = 0.048).

Seven potential risk factors were found to be associated with osteoporosis through univariate analysis: age, obesity, varices, grade of varices, Child–Pugh score, MELD score, and alanine aminotransferase (ALT). As in Table 3, multivariate analysis identified five significant factors associated with osteoporosis in patients with LC: age, obesity, grade of varices, Child–Pugh score, and MELD score.

## 4. Discussion

The results found in our study showed a superior prevalence of malnutrition (as evaluated with the CONUT score) for patients with a more advanced Child–Pugh level of cirrhosis. Moreover, taken separately, biochemistry markers such as high values of AST and total bilirubin corresponded to a more severe CONUT stage. The level of malnutrition was strongly correlated with liver function as determined by the MELD score. Moreover, the CONUT values were not outright related to QCT BMD data, but we found a good correlation between BMD and the presence of varices, grade of varices, age, obesity, Child–Pugh score, and MELD score, with all of them being statistically significant.

In patients with LC, it has been observed that bone loss is a frequent complication, one that can develop osteopenia levels up to 50% and osteoporosis levels ranging from 10 to 45% [32,33,34]. Among the mechanisms involved in the occurrence of bone disorders, hepatic cholestasis turned out to facilitate bone abnormalities in patients with LC, leading also to metabolic bone impairment and osteoporosis in later stages [35,36]. We also observed that high values of bilirubin and the presence of jaundice were matched up with higher stages of the CONUT index. The decrease in BMD in these patients can be caused by a set of etiopathogenic factors that mainly induce osteoblastic impairment, in spite of the fact that there also may be a certain level of osteoclastic hyperactivity [37,38]. Osteoporosis was detected in 24% to 38% of patients with end-stage disease in multiple etiologies [39,40,41]. Guichelaar et al. stated that osteoporosis was found in almost 38% of patients, osteopenia in 39%, but only 23% of patients had normal bone mass in a group of 360 patients with advanced cholestatic liver disease (primary biliary cholangitis and primary sclerosing cholangitis) [39,42]. In addition, patients with end-stage liver disease frequently suffer fractures [43]. QCT is useful in different stages of prediction, diagnosis, and prevention of osteoporosis and fractures [15,44]. Volumetric bone density measured by QCT is more sensitive to any modifications in BMD than DXA, which evaluates area bone density. Homologous findings by QCT measurement were also depicted in our study, in which low values of lumbar BMD were positively correlated with age and signs of portal hypertension such as presence of esophageal varices [17,45]. According with a previous study, our results demonstrated that the BMD measured by QCT was significantly lower in advanced stages of LC [46].

The present study attempted to identify the nutritional status among patients with LC with the use of the CONUT index in order to establish the association between malnutrition and bone loss. The results showed a correlation between a higher CONUT score and an advanced Child–Pugh classification of cirrhosis or biochemistry markers such as high values of AST and total bilirubin. A recent meta-analysis has shown that the nutritional status evaluated by the CONUT score was associated with prognosis of the liver disease [47]. Moreover, it has been found that the CONUT score was associated with the prognosis of various cancers and especially with HCC [16,47,48,49]. The frequency of malnutrition in patients with liver disease ranges between 10% and 100%, depending largely on the characteristics of the patients and the methods of nutritional assessment performed [50]. Since the CONUT index is based on derivatives on the laboratory data by using blood samples, we easily and objectively evaluated the nutritional status of the patients [18,19]. In line with the literature, we found that malnutrition can be seen in all clinical stages but is visualized more frequently in advanced stages of liver disease [51]. Accordingly, alcoholic liver disease is more frequently related with malnutrition [52]. The prevalence found in clinical trials is between 20% for patients with compensated LC and 100% in hospitalized patients with acute alcoholic hepatitis superimposed on cirrhosis [53,54]. 

Our study also found a correlation between obesity and low bone mass. This involves the fact that fat and bone mass impart some environmental factors, which can relate to the risk of osteoporosis [55]. A previous analytic study indicated that higher fat mass is related to lower BMD [56].

Even though evidence that low calcium contributes to osteoporosis development is weak, there is contradictory data for calcium abnormalities in LC. We noticed no differences in calcium levels between patients with alcoholic liver disease or virally induced liver cirrhosis [57,58].

This study has also confirmed that there is a direct positive correlation between T-score (BMD DEXA) and lumbar QCT. Furthermore, studies have confirmed that QCT is more efficacious than DEXA scan and that it also helps discriminate between groups of patients [59,60].

A higher CONUT score was related to an advanced Child–Pugh stage and MELD index with its laboratory variables being able to be considered warning signs regarding latter stages of bone disease such as osteopenia and osteoporosis [61].

Several limitations were found in the study and should be mentioned. The first reason encountered is that this is a retrospective observational study. The second one is linked to the idea that the study was based only on 70 patients diagnosed with viral B, viral C, or alcohol-induced LC cohort during pandemics, as well as additional studies on different liver diseases such as autoimmune are essentials to further confirm and extrapolate to other terms. From our thorough literature research, we encountered that there are no evident correlations between calcium levels and the severity of liver and bone disease [62]. Still, current outcomes from the research demonstrated that the CONUT score is well correlated with liver function and laboratory parameters such as Child–Pugh stage, MELD score, or AST and total bilirubin, which can be useful for predicting malnutrition as defined by the CONUT score.

## 5. Conclusions

Our findings showed a higher degree of malnutrition assessed with the CONUT index, considerably correlated with liver function as determined by the MELD score. We also found a greater prevalence of malnutrition among patients with LC. The CONUT values were not accurately related to QCT BMD data. Therefore, a strong correlation between BMD and the presence of varices, grade of varices, age, obesity, Child–Pugh score, and MELD score was found; moreover, all of them were statistically significant. Thus, bearing in mind that the CONUT has not been confirmed as a definitive marker of nutritional status in LC, assumptions about its link with liver function and bone density should be observantly drawn. Further research should contain more studies on the advantages and utility of the CONUT score for nutritional analysis in liver disease. There should also be more attention on verifying the BMD in opportunistic CT scan for osteoporosis assessment when diagnosing patients with LC.

## Figures and Tables

**Figure 1 jcm-12-00153-f001:**
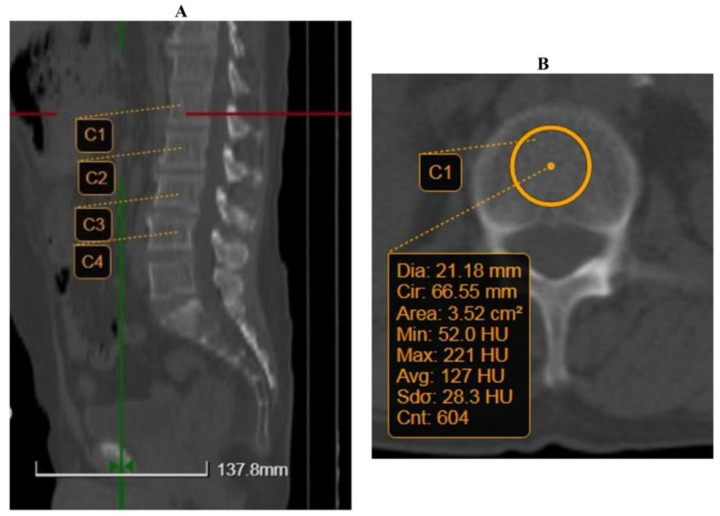
Example of quantitative computed tomography (QCT) measurement. (**A**) Reconstructions were obtained in the axial plane. (**B**) Region of interest (ROI) was chosen to be around 20 mm. C1–C4, L1–L4 lumbar vertebral corpus.

**Figure 2 jcm-12-00153-f002:**
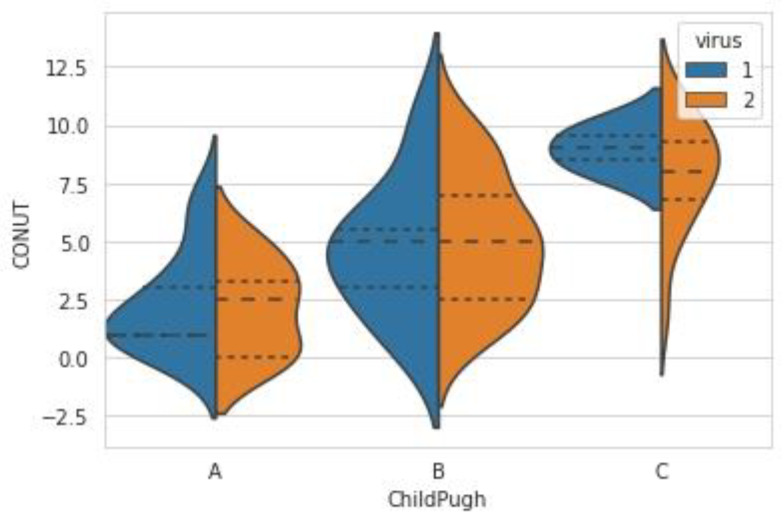
The CONUT scores according to Child–Pugh classification (1—virally induced liver cirrhosis, 2 = alcohol-induced liver cirrhosis).

**Figure 3 jcm-12-00153-f003:**
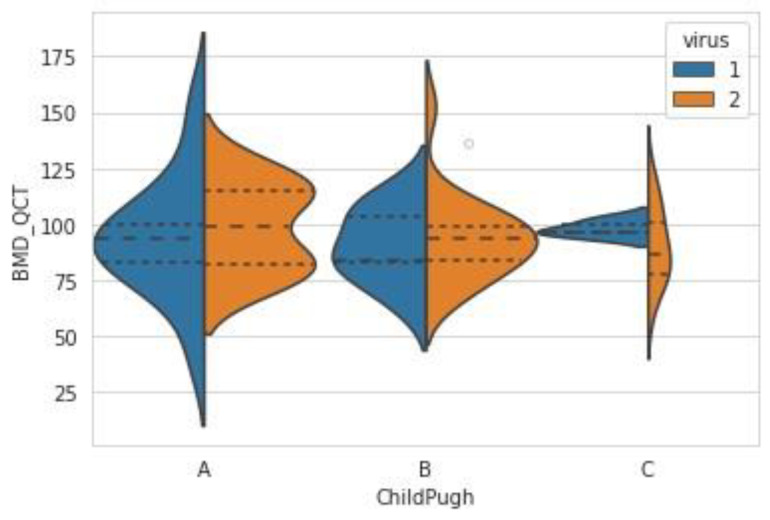
The BMD QCT values according to Child–Pugh classification (1—virally induced liver cirrhosis, 2 = alcohol-induced liver cirrhosis).

**Figure 4 jcm-12-00153-f004:**
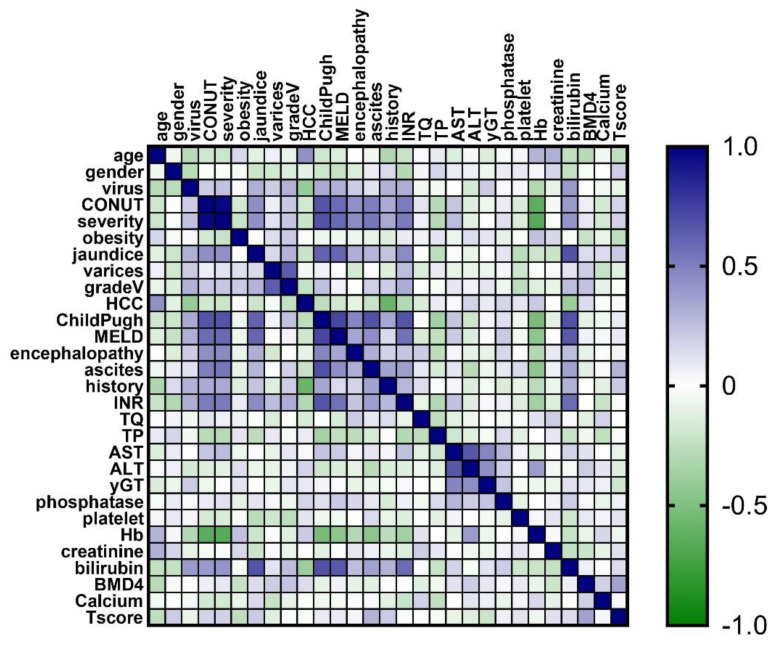
The correlation heatmap between measured indicators (colors range from bright blue for strong positive correlations to bright green for strong negative correlations). BMD, bone mineral density.

**Table 1 jcm-12-00153-t001:** Baseline data and comparison between viral and alcohol cirrhosis patients.

Parameters	Total Patients n = 70	Virally Induced LC (n = 23)	Alcohol-Induced LC (n = 47)	*p*-Value
Age, years	59.8 (±10.8) 62 (53–66.5)	64.57 (±10.25) 64 (57–72)	57.47 (±10.38) 61 (51–65)	0.020
Gender, male	65 (92.9%)	19 (82.61%)	46 (97.87%)	0.037
History of hepatic decompensation, yes	55 (78.6%)	14 (60.87%)	41 (87.23%)	0.015
Encephalopathy, yes	14 (20%)	2 (8.7%)	12 (25.53%)	0.087
Ascites, yes	33 (47.1%)	9 (39.13%)	24 (51.06%)	0.247
CONUT score	4.86 (±3.28) 5 (2–8)	3.96 (±3.17) 3 (1–6)	5.3 (±3.27) 5 (2–8)	0.101
Severity				0.158
Normal	14 (20%)	8 (34.78%)	6 (12.77%)	
Mild	18 (25.7%)	6 (26.09%)	12 (25.53%)	
Moderate	25 (35.7%)	6 (26.09%)	19 (40.43%)	
Severe	13 (18.6%)	3 (13.04%)	10 (21.28%)	
Obesity				0.519
Underweight	5 (7.1%)	3 (13.04%)	2 (4.26%)	
Normal	35 (50%)	10 (43.48%)	25 (53.19%)	
Overweight	20 (28.6%)	6 (26.09%)	14 (29.79%)	
Obese	10 (14.3%)	4 (17.39%)	6 (12.77%)	
Jaundice, yes	24 (34.3%)	3 (13.04%)	21 (44.68%)	0.008
Cirrhosis, yes	70 (100%)	23 (100%)	47 (100%)	-
Varices, yes	59 (84.3%)	17 (73.91%)	42 (89.36%)	0.096
Varices grade				0.142
0	13 (18.6%)	6 (26.09%)	7 (14.89%)	
1	19 (27.1%)	9 (39.13%)	10 (21.28%)	
2	25 (35.7%)	7 (30.43%)	18 (38.30%)	
3	12 (17.1%)	1 (4.35%)	11 (23.40%	
4	1 (1.4%)	0	1 (2.13%)	
Child–Pugh				0.029
A	25 (35.7%)	13 (56.52%)	12 (25.53%)	
B	26 (37.1%)	7 (30.43%)	19 (40.43%)	
C	19 (27.1%)	3 (13.04%)	16 (34.04%)	
MELD	15.59 (±6.45) 15 (10–19.25)	12.96 (±5.66) 10 (8–18)	16.87 (±6.48) 16 (11–21)	0.011
INR	1.43 (±0.38) 1.32 (1.12–1.66)	1.3 (±0.4) 1.13 (1.11–1.35)	1.49 (±0.36) 1.42 (1.18–1.7)	0.008
TQ	28.45 (±10.35) 30.8 (18–36)	29.41 (±9.98) 32.8 (18–36)	27.98 (±10.6) 30.6 (18–35.8)	0.608
TP	62.57 (±22.3) 59.5 (46.7–78.22)	62.87 (±25.37) 61 (45.8–83)	62.43 (±20.93) 57 (47–70)	0.657
AST (U/L)	61.89 (±39.06) 51 (36.75–74.75)	58.96 (±30.88) 49 (40–74)	63.33 (±42.74) 52 (36–82)	0.995
ALT (U/L)	35.84 (±23.49) 29.5 (19–47)	42.61 (±28.27) 34 (19–57)	32.53 (±20.27) 27 (19–44)	0.163
yGT	215.64 (±341.19) 93 (41–213)	88.59 (±62.05) 83.5 (37–124.75)	275.11 (±398.74) 102 (47–323)	0.094
Phosphatase	116.66 (±53.87) 107.5 (72.75–136.5)	110 (±39.96) 110 (73–128)	119.91 (±59.63) 103 (71–158)	0.851
Calcium (mmol/L)	7.93 (±0.29) 7.93	7.95 (±0.2) 7.93	7.92 (±0.33) 7.93	0.690
Platelet count (10^9^/L)	138.5 (±83.92) 123.6 (84.58–179.95)	138.44 (±93.74) 123 (85.44–166)	138.53 (±79.76) 124.2 (81.99–188.2)	0.812
Hemoglobin (g/dL)	11.22 (±2.92) 11.76 (8.75–13.65)	12.47 (±2.85) 12.88 (10.04–15.39)	10.6 (±2.78) 10.48 (8.41–12.96)	0.012
Creatinine (mg/dL)	0.93 (±0.51) 0.8 (0.69–0.99)	1.05 (±0.76) 0.86 (0.7–1.07)	0.88 (±0.33) 0.79 (0.69–0.92)	0.445
Bilirubin (mg/dL)	3.07 (±3.89) 1.72 (1.28–2.68)	1.6 (±1.17) 1.33 (0.86–1.69)	3.79 (±4.52) 2.1 (1.53–3.12)	0.001

Abbreviations: Data are expressed in mean (±SD), median (interquartile range), or number (percentage). LC, Liver Cirrhosis; MELD, Model for End-Stage Liver Disease score; INR, International Normalized Ratio; TQ, Time of Quick; TP, Prothrombin Time; AST, aspartate aminotransferase; ALT, alanine aminotransferase; yGT, gamma-glutamyltransferase.

**Table 2 jcm-12-00153-t002:** Comparative scores for QCT and osteoporosis.

	Total Patients n = 70	Virally Induced LC (n = 23)	Alcohol-Induced LC (n = 47)	*p*-Value
QCT score	92.18 (±17.86) 88.41 (81.68–104.57)	89.93 (±17.68) 87.53 (82.74–102.15)	93.28 (±18.03) 88.66 (79.65–108.07)	0.684
T-score	−1.68 (±0.62) −1.68	−1.7 (±0.73) −1.68	−1.67 (±0.56) −1.68	0.493
Osteoporosis				0.870
Normal	6 (5.7%)	2 (8.7%)	4 (8.5%)	
Osteopenia	49 (70%)	17 (73.91%)	32 (68.1%)	
Osteoporosis	15 (24.3%)	4 (17.4%)	11 (23.4%)	

Abbreviations: Data are expressed as mean (±SD), median (interquartile range), or number (percentage). QCT, quantitative computed tomography.

**Table 3 jcm-12-00153-t003:** Significant factors in the multivariate analysis for osteoporosis in liver cirrhosis patients.

Variables	Multivariate Analysis		
	Odds Ratio	95% Confidence Interval	*p*-Value
Age	−0.45	−0.74 to −0.06	0.025
Obesity	−6.67	−11.73 to −1.6	0.011
Varices	−6.53	−21.63 to 8.6	0.391
Grade of varices	7.69	1.89 to 13.49	0.010
Child–Pugh	−4.33	−7.16 to −1.51	0.003
MELD	1.2	0.23 to 2.17	0.016
ALT	0.14	−0.03 to 0.32	0.109

Abbreviations: MELD, Model for End-Stage Liver Disease score; ALT, alanine aminotransferase.

## Data Availability

The data used to support the findings of this study are available from the corresponding author upon reasonable request.

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
