# Peer review of "Osteoporosis Assessment among Adults with Liver Cirrhosis"

_jcm, 2022, doi:10.3390/jcm12010153_

Round 1
Reviewer 1 Report
This study correlates BMD levels measured with lumbar QCT to CONUT-assesed nutritional status in various forms of liver cirrhosis. Although the hepatological aspect is well described and argued, I don't think the same can be said for osteoporosis, complication analyzed retrospectively in this cohort of patients. I think that to conduct a correct correlation between bone mass and nutritional status in these patients, it is not sufficient to evaluate only the BMD measured with QCT, because BMD can vary for different factors that need to be described. In my opinion there are numerous missing data that do not allow an adequate discussion of the results. There are several points that should be analysed:
- T score and Z score of lumbar QCT?
- the population should be divided into: post menopausal and pre menopausal females/males (are there any differences? considering the different effects of sex and menopausal status on BMD)
- use of drugs interfering with bone metabolism?
- BMI score?
- history of fragility fractures?
- even if the BMD DEXA has limitations, it could be useful to compare the BMD DEXA data with the lumbar QCT
- I understand the retrospective nature of the study and its intrinsic limitations, but it is difficult to make an assessment of bone mass related to liver disease, without any data relating to mineral metabolism (calcemia, phosphataemia, bone alkaline phosphatase, 24-hour calciuria, PTH.. ..)
- there is a lack of data on 25oh vitamin D levels, knowing well the importance of vitamin D for mineralization and knowing that in these patients there may be a reduction in the activity of hepatic 25hydroxylase
- kidney function?
- did the patients have no other pathologies that could cause alterations in BMD?
I do not think that this study can be accepted because other factors potentially interfering with BMD have not been evaluated.
Author Response
We are very grateful to the constructive comments from you. We also thank you for the time and effort on reviewing our manuscript.
We have carefully addressed point-by-point all the comments and made corrections in our manuscript using tracked changes.
- - T score and Z score of lumbar QCT?
We appreciate very much your question. In order to facilitate the interpretation of QCT spine results, we used the criteria of American College of Radiology which were introduced in 2008 and 2013 and revised in 2018 published guidelines for the performance of QCT; based on these guidelines, trabecular BMD values from 120 mg/cm3 to 80 mg/cm3 are defined as osteopenia and BMD values less than 80 mg/cm3 as osteoporosis (1). The use of T-scores has been avoided in this categorization to reinforce the fact that QCT spine T-scores and hip T-scores are frequently different. (2)
- - the population should be divided into: post menopausal and pre menopausal females/males (are there any differences? considering the different effects of sex and menopausal status on BMD)
Thank you for your comment. Taking in count the retrospective aspect of the study, there weren’t any data about premenopausal or postmenopausal status of the 5 women which met the inclusion criteria. We analyzed the effect of gender on BMD and no differences were found between males and females (p-value =0.732). Because of the small number of females, we evaluated the power test and the achieved power was 5.57% (two-tails, α = 0.05) and we did not include these results into the manuscript.
- - use of drugs interfering with bone metabolism?
At your recommendation we modified the inclusion/exclusion criteria by rechecking all the patient’s admission files in the hospital. All the 70 patients did not take any drugs or oral supplements containing magnesium, calcium, vitamin D and phosphorus. Also, we checked the viral B patients, but there were not treated by diuretics and Tenofovir disoproxil (TDF), only Entecavir.
It is known that patients with decompensated liver cirrhosis are taking diuretics and Dohyeong et al suggested that liver function can influence the incidence of hypophosphatemia, and the risk of hypophosphatemia in patients with cirrhosis is about 3.4-fold greater than in patients with normal liver function. Other drugs or oral supplements like magnesium, calcium, vitamin D or phosphorus were checked and none of those patients took before admission (3).
We modified the exclusion criteria from the study with: patients who were under treatment with calcium, magnesium or phosphorus supplements. Also, we did not take into consideration for this study patients diagnosed with auto-immune liver diseases, hepatocellular carcinoma or other malignancies, kidney failure and documented bone disease like osteopenia and osteoporosis. If fractures or past fractures, cement augmented vertebral bodies, anatomical deformities, or implants were detected in the CT scan, the respective patients were not included in the measurements.
- - BMI score?
Thank you for the question. We extensively reevaluated the patients documents and we did not find any information about BMI score, only the level of obesity (underweight, normal, overweight or obese).
- - history of fragility fractures?
Thank you for the opportunity to ask this important question. All the patients were asked about the history of fragility fractures when signed the informed consent and none of them had.
- - even if the BMD DEXA has limitations, it could be useful to compare the BMD DEXA data with the lumbar QCT
We thank for the comment. We carefully checked the patient’s hospitalization files and we mentioned the new results in the revised version of the manuscript. As we had missing data for calcium and t-score (BMD DEXA), we considered the outcomes could be skewed, with low statistical power of the analysis, distorting the validity of the results.
The data are not missing by design, but because of the retrospective type of the study, and the statistical advantage of this type of missing data (missing completely at random) is that the analysis remains unbiased. (4)
Because of the small sample, the listwise or pairwise deletion was not the optimal strategy. We used mean substitution, increasing data with reduced standard error.
We obtained a positive significant correlation between BMD DEXA and lumbar QCT (Spearman's rho = 0.361, p-value = 0.002).
- - I understand the retrospective nature of the study and its intrinsic limitations, but it is difficult to make an assessment of bone mass related to liver disease, without any data relating to mineral metabolism (calcemia, phosphataemia, bone alkaline phosphatase, 24-hour calciuria, PTH.. ..)
We introduced the new results into the manuscript after we assessed calcemia with the new missing data design.
- - there is a lack of data on 25oh vitamin D levels, knowing well the importance of vitamin D for mineralization and knowing that in these patients there may be a reduction in the activity of hepatic 25hydroxylase
Thank you very much for this important comment. Taking into count that our study was retrospective, we don’t have any data about vitamin D in our patients. Currently, vitamin D isn’t checked by default in cirrhotic patients but we would like to further expand our research in this direction.
- - kidney function?
Thank for this important question which can better orientate the reader. All the patients that met the inclusion criteria were with normal kidney function, none of our patients had end stage kidney disease.
- - did the patients have no other pathologies that could cause alterations in BMD?
It is an important subject of discussion and we modified the inclusion/exclusion criteria and double checked them. We really appreciate your interest in reviewing our study. After an extensive analysis of the patient’s files, we found the following comorbidities: arterial hypertension and other cardiovascular diseases, but without medication influencing BMD.

Reviewer 2 Report
Dear author:
Regarding your abstract entitled " Osteoporosis assessment among adults with multiple etiologies of liver cirrhosis " which studied the relationship between nutritional status and bone mineral density in liver cirrhosis. I appreciate your nice work but, I have a few concerns before the final decision.
- Language edition is required.
- Remove abbreviations from the abstract part.
- In your study you included only patient with liver cirrhosis due to alcoholic and viral hepatitis so I think the title of the abstract should be changed " Osteoporosis assessment among adults with liver cirrhosis"
- Could you explain why you excluded those with autoimmune hepatitis and others.
- Sample size calculation
- What do you mean by "hepatic decompensation"? p value was positive despite there was no differences between the two group as regard ascites, encephalopathy and varices
I think it would be better if regression analysis was done
Author Response
The authors would like to thank the reviewer for the suggestion to improve the introduction, new references have been inserted in the text of the paper and the introduction was reworded. We have studied all referees’ comments carefully and have made corrections using tracked changes.
- Language edition is required.
Thank you for the suggestion, all the manuscript was checked and rephrased where needed by a native English speaker.
- Remove abbreviations from the abstract part.
Thank you very much for this important comment about the abstract. We removed the abbreviations as following:
Osteopenic bone disease occurs frequently in patients with chronic liver cirrhosis, which commonly present with hepatic osteodystrophy. Thus, the relationship between nutritional status and bone mineral density has been poorly studied in liver cirrhosis. This single-center study included a group of 70 patients diagnosed with liver cirrhosis. The nutritional status was assessed with the Controlling Nutritional Status index; volumetric vertebral bone mineral density measured with quantitative computed tomography. Among the 70 patients included, osteopenia and osteoporosis were present in 71% and 24.3%, respectively. Malnutrition estimated with the Controlling Nutritional Status index was observed in 56 (80%) of patients and was more frequent in alcoholic cirrhosis patients than viral cirrhosis patients (87.24% vs. 65.22%). Significant positive correlation with Controlling Nutritional Status score was found with Model for End-Stage Liver Disease (rho = 0.576, p-value < 0.0001), Child-Pugh (rho = 0.670, p-value < 0.0001), International Normalized Ratio (rho = 0.517, p-value = 0.001), aspartate aminotransferase (rho = 0.293, p-value = 0.045), and bilirubin (rho =0.395, p-value = 0.02). Among the liver cirrhosis patients, 15 had osteoporosis and 49 had osteopenia at the lumbar spine (L1-L4 vertebrae) as determined by bone mass density via quantitative computed tomography. A non-significant relationship between Controlling Nutritional Status index-assessed nutritional status and bone mass density was documented. Regarding osteoporosis, no differences were found between viral and alcohol types of liver cirrhosis patients (p-value = 0.870).
- In your study, you included only patient with liver cirrhosis due to alcoholic and viral hepatitis so I think the title of the abstract should be changed "Osteoporosis assessment among adults with liver cirrhosis"
Thank you very much for this important comment. We modified the title of the manuscript as following: “Osteoporosis assessment among adults with liver cirrhosis”
- Could you explain why you excluded those with autoimmune hepatitis and others.
We appreciate very much your question. The reason that we excluded the patients with autoimmune liver disease was the lack of data. In the period of 2019-2020, there were only 3 patients with primary sclerosing cholangitis and 3 patients with primary biliary cholangitis but all of them without computed tomography performed.
- Sample size calculation
The power analysis for our study was performed using G*Power 3.1.9.7, at a 95% confidence level and power factor of 80% for each of the groups. A two-sided p-value smaller than 0.05 was considered to be statistically significant. The power test was done and assuming an alpha level of 0.05, the patients from viral induced liver cirrhosis and alcohol induced liver cirrhosis groups yielded the power between 67% and 86% for the different analysis.
- What do you mean by "hepatic decompensation"? p value was positive despite there was no differences between the two group as regard ascites, encephalopathy and varices
Thank again for this important question. We tried to find if there are differences between the two etiologies and we concluded that only jaundice was more frequent in patients diagnosed with alcoholic liver cirrhosis. Table 1 presents the differences between history of hepatic decompensation. The other forms of hepatic decompensation like presence of ascitic liquid, encephalopathy and esophageal varices were found in similar proportions in the two groups of patients after the treatment. As outcomes after treatment with beta-blockers and after banding sessions of the esophageal varices, all the patients were in remission (1). Ascitic liquid was removed from the abdomen through a slender needle and all the patients responded after paracentesis.
- I think it would be better if regression analysis was done
The regression analysis was done as we presented in Methods section (line 533-536) and Results section (line 619-626).
References
- Rodrigues SG, Mendoza YP, Bosch J. Beta-blockers in cirrhosis: Evidence-based indications and limitations. JHEP Reports. 2020 Feb 1;2(1).
Round 2
Reviewer 1 Report
I think the authors reviewed the article properly, so the article is now acceptable for publication.